# Alternative poly-adenylation modulates α1-antitrypsin expression in chronic obstructive pulmonary disease

Lela Lackey[1]*, Aaztli Coria[2,3], Auyon J. Ghosh[4,5], Phil Grayeski[6], Abigail Hatfield[1], Vijay Shankar[1], John Platig[4], Zhonghui Xu[4], Silvia B. V. Ramos[3], Edwin K. Silverman[4,5], Victor E. Ortega[7], Michael H. Cho[4,5], Craig P. Hersh[4,5], Brian D. Hobbs[4,5], Peter Castaldi[4,8], Alain Laederach[2]*

1 Department of Genetics and Biochemistry, Center for Human Genetics, Clemson University, Greenwood, South Carolina, United States of America, 2 Department of Biology, University of North Carolina, Chapel Hill, North Carolina, United States of America, 3 Department of Biochemistry and Biophysics, University of North Carolina, Chapel Hill, North Carolina, United States of America, 4 Channing Division of Network Medicine, Brigham and Women's Hospital, Harvard Medical School, Boston, Massachusetts, United States of America, 5 Division of Pulmonary and Critical Care Medicine, Brigham and Women's Hospital, Harvard Medical School, Boston, Massachusetts, United States of America, 6 Curriculum in Genetics and Molecular Biology, University of North Carolina, Chapel Hill, North Carolina, United States of America, 7 Department of Internal Medicine, Division of Respiratory Medicine, Center for Individualized Medicine, Mayo Clinic, Scottsdale, Arizona, United States of America, 8 Division of Internal Medicine and Primary Care, Brigham and Women's Hospital, Harvard Medical School, Boston, Massachusetts, United States of America

* lelal@clemson.edu (LL); alain@unc.edu (AL)

**Data Availability Statement:** The data underlying the results presented in the study are available from the Sequence Read Archive (SRA) and database of Genotypes and Phenotypes (dbGAP).

## Abstract

α1-anti-trypsin (A1AT), encoded by *SERPINA1*, is a neutrophil elastase inhibitor that controls the inflammatory response in the lung. Severe A1AT deficiency increases risk for Chronic Obstructive Pulmonary Disease (COPD), however, the role of A1AT in COPD in non-deficient individuals is not well known. We identify a 2.1-fold increase (p = $2.5 \times 10^{-6}$) in the use of a distal poly-adenylation site in primary lung tissue RNA-seq in 82 COPD cases when compared to 64 controls and replicate this in an independent study of 376 COPD and 267 controls. This alternative polyadenylation event involves two sites, a proximal and distal site, 61 and 1683 nucleotides downstream of the A1AT stop codon. To characterize this event, we measured the distal ratio in human primary tissue short read RNA-seq data and corroborated our results with long read RNA-seq data. Integrating these results with 3' end RNA-seq and nanoluciferase reporter assay experiments we show that use of the distal site yields mRNA transcripts with over 50-fold decreased translation efficiency and A1AT expression. We identified seven RNA binding proteins using enhanced CrossLinking and ImmunoPrecipitation precipitation (eCLIP) with one or more binding sites in the *SERPINA1* 3' UTR. We combined these data with measurements of the distal ratio in shRNA knockdown experiments, nuclear and cytoplasmic fractionation, and chemical RNA structure probing. We identify Quaking Homolog (QKI) as a modulator of *SERPINA1* mRNA translation and confirm the role of QKI in *SERPINA1* translation with luciferase reporter assays. Analysis of single-cell RNA-seq showed differences in the distribution of the *SERPINA1* distal ratio among hepatocytes, macrophages, αβ-Tcells and plasma cells in the liver. Alveolar Type 1,2, dendritic cells and macrophages also vary in their distal ratio in the lung. Our work

Datasets used in this study include the GTEX project (dbGaP Accession phs000424.v8.p2), NHLBI TOPMed: Lung Tissue Research Consortium (dbGAP Accession phs001662.v2.p1), lung RNA-seq data (Bioproject PRJNA245811), ENCODE HepG2 fractionation experiments (Bioproject PRJNA30709:GSE30567 - SRR307915, SRR307916, SRR307928, SRR307929), hepatocellular carcinoma cell line fractionation experiments (Bioproject PRJNA543441) and long read PacBio human liver IsoSeq alignments (https://s3.amazonaws.com/datasets.pacb.com/downloadtools.html).

**Funding:** This work was supported by National Institutes of Health, National Institute of General Medical Sciences (NIGMS) grant R35 GM140844 (https://www.nigms.nih.gov/) and National Institute of Heart, Lung and Blood (NHLBI) (https://www.nhlbi.nih.gov/) R01 HL111527 to AL and NIGMS grant R35 GM142851 and an Alpha-1 Foundation post-doctoral fellowship 615028 (https://www.alpha1.org/) to LL. Additional support was provided by NHLBI grants R01 HL124233 and R01 HL147326 to PJC and K25 HL140186 to JP and K08 HL136928 to BDH. Molecular data for the Trans-Omics in Precision Medicine (TOPMed) program was supported by the NHLBI. This study utilized biological specimens and data provided by the Lung Tissue Research Consortium (LTRC) supported by the NHLBI. Whole Genome Sequencing and RNASeq for NHLBI TOPMed: The Lung Tissue Research Consortium (phs001662) was performed at the Broad Institute Genomics Platform (HHSN2682016000034I) and Northwest Genomics Center (HHSN2682016000032I). Core support including centralized genomic read mapping and genotype calling, along with variant quality metrics and filtering were provided by the TOPMed Informatics Research Center (3R01HL-117626-02S1; contract HHSN268201800002I). Core support including phenotype harmonization, data management, sample-identity QC, and general program coordination were provided by the TOPMed Data Coordinating Center (R01HL-120393; U01HL-120393; contract HHSN268201800001I). The funders had no role in study design, data collection and analysis, decision to publish, or preparation of this manuscript.

**Competing interests:** We have read the journal's policy and the authors of this manuscript have the following competing interests: EKS received grant support from GlaxoSmithKline and Bayer. MHC has received grant support from GlaxoSmithKline and Bayer, consulting fees from Genentech and AstraZeneca, and speaking fees from Illumina. CPH reports grant support from Boehringer-Ingelheim,

reveals a complex post-transcriptional mechanism that regulates alternative polyadenylation and A1AT expression in COPD.

## Author summary

In certain cases, the molecular mechanism underlying a human genetic disease association occurs at the level of the RNA transcript. The *SERPINA1* gene is transcribed into multiple RNA isoforms that differ in their non-coding regions and influence production of the A1AT protein, which is associated with Chronic Obstructive Pulmonary Disease (COPD). Here we report a newly discovered alternative polyadenylation event as an underlying regulatory mechanism that influences A1AT protein expression. In the *SERPINA1* mRNA there are two polyadenylation sites in the 3' untranslated region (3'UTR), resulting in a short and a long transcript. We find that the longer 3'UTR strongly represses protein production, and is expressed at a higher level in individuals with COPD, in two independent studies of lung tissue. The inhibitory mechanism in the long transcript is likely caused by an interaction with the RNA binding protein Quaking, which is also differentially expressed in individuals with COPD. This work demonstrates how post-transcriptional regulation can affect disease and also identifies a potential target in the SERPINA1 3' UTR for RNA targeted therapeutics.

## Introduction

The α1-anti-trypsin (A1AT) protein functions primarily as a neutrophil elastase inhibitor and controls the inflammatory response in the lung [1,2]. It is encoded in the *SERPINA1* messenger RNA (mRNA), that is translated into a single protein isoform [3]. Despite coding for a single protein isoform, the *SERPINA1* mRNA is spliced into 11 transcript variants, with all variants differing only in their 5' UnTranslated Region (UTR). This puts the gene in the top 5% of most transcriptionally complex genes in the human genome [4]. Interestingly, these 5' UTR transcript variants selectively include and exclude upstream Open Reading Frames (uORFs) which regulate the translation efficiency of the mRNA ultimately affecting A1AT expression [4,5]. Recent deep resequencing of the *SERPINA1* locus in the SubPopulations and InteRmediate Outcome Measures In COPD Study (SPIROMICS) identified additional variation mapping to the 5' UTR of the gene associated with lowered A1AT serum levels and functional small airway disease [6]. These results support that post-transcriptional regulation of expression is an important and understudied component of A1AT function and lung physiology.

A1AT deficiency is a hereditary disorder that can lead to panlobular emphysema in the lung and cirrhosis in the liver [7–9]. A1AT is secreted and predominantly produced by the liver [7,8]. Accordingly, *SERPINA1* mRNA is most highly expressed in the liver, however there is also significant expression in blood, lung, small intestine, spleen and kidney in humans (Fig 1A) [10,11]. Although the role of the A1AT protein in disease etiology is well established, the *SERPINA1* mRNA, and in particular its multiple alternative splicing transcriptional isoforms is exceptional and still poorly understood [3]. To date, the role of the 3' UTR of the *SERPINA1* mRNA in A1AT expression has yet to be investigated. 3' UTRs of mRNAs are known to control both stability and translation efficiency of the message through multiple mechanisms

Novartis, Bayer, Vertex, and personal fees from
Takeda outside of this study. PJC has received
grant support from GlaxoSmithKline and Bayer and
consulting fees from GlaxoSmithKline and
Novartis. VEO received fees for participation in
independent data and monitoring committees for
Regeneron and Sanofi and consulting fees from
Sanofi. AL has received consulting fees from
Ribometrix. PJG holds equity in Ribometrix, to
which correlated chemical probing technologies
have been licensed. LL, AC, AJG, AH, VS, JP, ZX,
SBVR, and BDH have no conflicts of interest to
report.

including RNA interference, RNA Binding Protein (RBP) binding, and alternative polyadenylation [12–16].

One hallmark of both A1AT protein and *SERPINA1* mRNA expression is significant variability between individuals [6]. However, this variability is not clearly associated with lung function, particularly when accounting for the inflammatory status as measured by C-reactive protein [17]. Although some of this variability can be attributed to genetic factors, such as the Z allele which causes accumulation of A1AT in the liver, the majority cannot be accounted for by *cis* genetic variation alone [7,8]. We investigate here alternative 3′ polyadenylation in the *SERPINA1* mRNA and its role in Chronic Obstructive Pulmonary Disease (COPD), a leading cause of death in the world [18–20]. We characterize a distal polyadenylation site in the *SERPINA1* mRNA that is differentially used in the lungs of COPD individuals. By combining quantitative 3′ end sequencing, large scale transcriptomic profiling of individuals, quantitative nanoluciferase reporter assays, RNA chemical structure probing, and single-cell RNA sequencing, we characterize the role of *SERPINA1* Alternative Poly-Adenylation (APA) in controlling A1AT translation. In addition, we identify key RNA binding proteins affecting the mechanism of A1AT post-transcriptional regulation. This novel mechanism is significant because it reveals a complex post-transcriptional control mechanism affecting A1AT expression that is an important component of A1AT deficiency disease etiology and COPD.

## Results

### Tissue specific expression of *SERPINA1* mRNA

We begin our investigation of post-transcriptional regulation by visualizing median expression of the *SERPINA1* mRNA across individuals sequenced in the Gene Tissue Expression (GTEx) Atlas (Fig 1A). We identified six tissues (Liver, Blood, Lung, Small Intestine, Spleen and Kidney) where the median expression was above 1000 TPM and therefore sufficient to obtain individual estimates of exon specific expression. In Fig 1B, we illustrate the standard deviation of expression across individuals in the GTEx data and map them to the human body [6,21]. Analysis of mean short-read coverage across individuals for these six tissues (Fig 1C) reveals well defined exons in the coding region of the gene (exons 4–7); the hg38 reference annotation of exons is shown in gray (untranslated) and black (protein coding) in Fig 1C. We observe differential coverage in exons 1–3 consistent with the tissue specific expression of *SERPINA1* mRNA transcript isoforms previously described [3,4,22,23].

The premise for our study is illustrated in Fig 1D where we have expanded exon 7, which includes the 3′ UTR of the mRNA. We observe a significant drop off in coverage approximately ~60 nucleotides downstream of the stop codon, which suggests the presence of distal and proximal alternative polyadenylation (APA) sites. The steep drop is also observed in coverage in RNA-seq data from Hep-G2 cell lines (Fig 1C, turquoise tracks). We therefore performed 3′-seq sequencing on these cells to quantitatively map the alternative poly-adenylation sites with single nucleotide resolution (Fig 1D). We observe two peaks in this signal (labeled proximal and distal) with sharp 3′ cliffs confirming that *SERPINA1* mRNA is alternatively poly adenylated. These peaks are consistent with the drop off in coverage in RNA-seq. Upon analysis of the sequence directly upstream of the distal drop off site, we observe a canonical AAUAAA motif consistent with a poly-A signal, while a weaker signal (AUUAAA) is present at the proximal site (S1A Fig). Furthermore our 3′-seq data allow us to map the cleavage site and identify the G/U-rich region downstream (shown in S1A Fig). These sequence motifs are hallmarks of APA in 3′ UTRs [11,24]. Finally, IGV analysis of PacBio long read RNA sequencing from primary human liver tissue confirms the existence of both long and short isoforms in the 3′ UTR as well as expression of multiple 5' UTR splice isoforms (Fig 1E) [25].

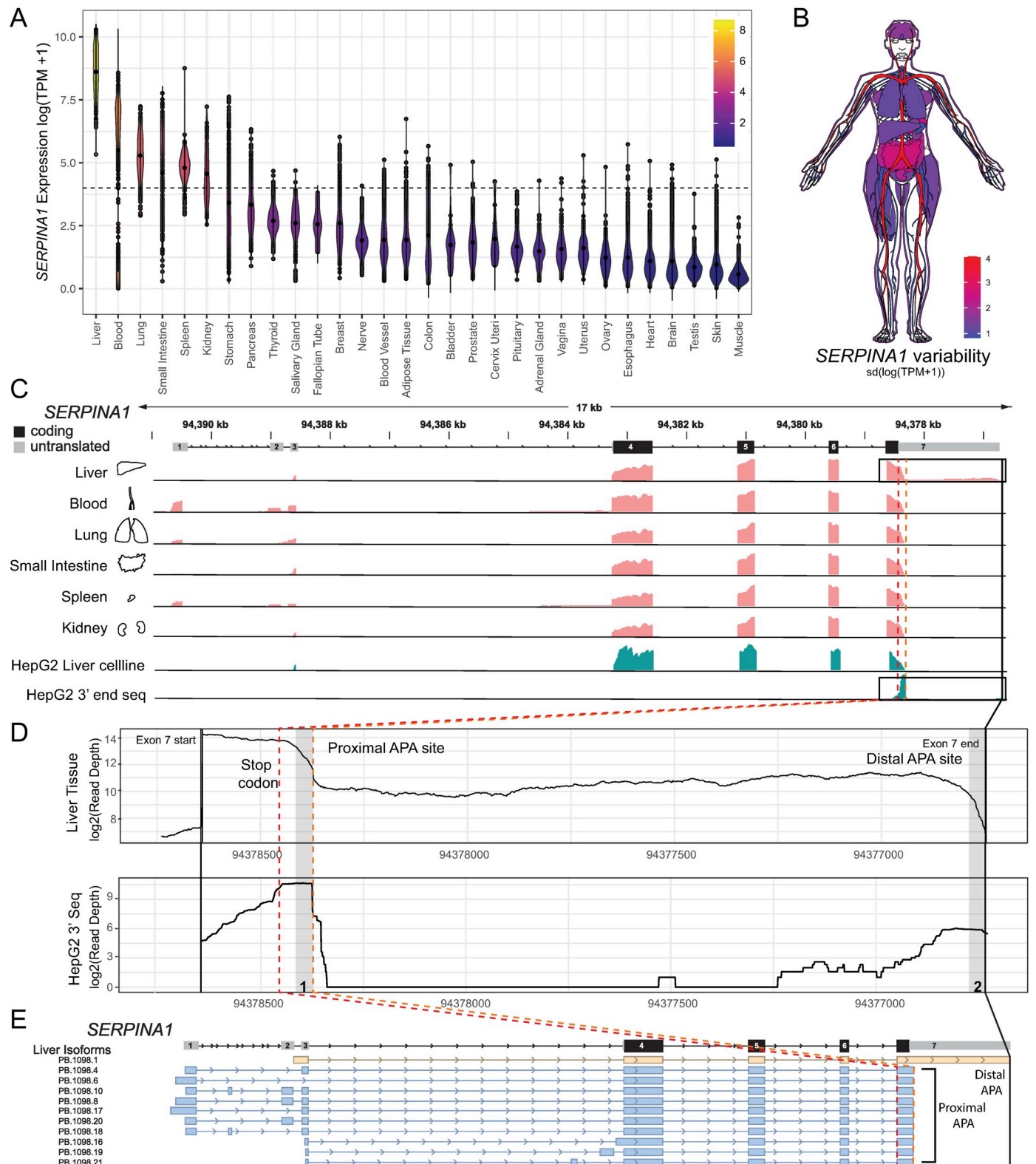

**Fig 1. Tissue expression and alternative polyadenylation of the *SERPINA1* mRNA.** A) Tissue expression in Transcripts per Million (TPM) on a log scale for *SERPINA1* mRNA for 17382 samples sequenced by the GTEx RNA-seq consortia [10,11,62]. The six tissues with median *SERPINA1* expression above average (~1000 TPM, indicated with dashed line) were further analyzed. B) Variation (as measured by the tissue specific standard deviation) in tissues. We observe particularly high levels of variation among individuals within blood samples. C) Read coverage of GTEx RNA-seq data for the six tissues highly expressing *SERPINA1* mRNA are shown

in pink. Read coverage for HepG2 and HepG2 3′ RNA-seq experiments shown in turquoise. All these RNA-seq data sets confirm the exon structure of *SERPINA1* as shown on top in black and grey. We observe tissue specific differential exon usage in the 5' UTR (exons 1–3) consistent with previous work characterizing differential isoform usage in this region of *SERPINA1* [4]. D) Zoom of the 3′ UTR region for primary liver data mean depth (top) and HepG2 3′end seq experiment identifying the proximal and distal APA sites. E) PacBio long read sequencing of primary human liver tissue data confirming isoforms using proximal (blue) and distal (yellow) APA sites.

Our results so far establish that the 3′ UTR of *SERPINA1* is alternatively polyadenylated, and that in all tissues (and HepG2 cell lines) the shorter isoform is preferentially expressed. Importantly, however, in all tissues and cell lines we observe coverage over the entire long isoform, suggesting that both isoforms are constitutively present, albeit at different levels. From the RNA-seq coverage data we compute the distal ratio for the six tissues by dividing the mean distal coverage depth by the proximal depth normalized by length. When we compute the distal ratio in human primary tissues, we observe the highest distal site usage in the liver (Fig 2A). Furthermore, we also observe significant variability in the distal ratio among individuals, especially in the lung and liver.

## APA in the lung and COPD

To investigate the role of APA in the *SERPINA1* 3′ UTR and its relationship with COPD we computed distal ratio in lung tissue from two independent population studies. In the first we analyzed a publicly available [26] short-read lung tissue RNA-seq data set from n = 82 subjects and n = 64 controls and observed a 2.1 (p = $2.5 \times 10^{-6}$) increase of the median distal ratio in individuals with COPD (Fig 2B). We therefore sought to replicate this finding in the larger Lung Tissue Research Consortium (LTRC) study where we obtained RNA-seq from 376 COPD cases and 267 controls (Fig 2C). There we observed a similar 1.5-fold (p < $7.6 \times 10^{-10}$) increase in the distal ratio, suggesting higher use of the distal polyadenylation site in individuals with COPD. This is also visualized as the mean normalized read depth for the LTRC data in Fig 2D. We observe in the LTRC data the characteristic drop-off at the proximal APA site consistent with our previous analysis of primary human tissues (Fig 1D).

The most common genetic variation leading to A1AT deficiency is the *SERPINA1* Z allele [7–9]. Individuals homozygous for the S allele (SS) and heterozygous for the Z allele (MZ) may also have reduced A1AT [23,27]. Although there are many other *SERPINA1* variant alleles, most are less common or do not significantly associate with A1AT levels [6]. In addition to this genetic effect, the likelihood of an individual to have lung-related A1AT deficiency-associated diseases such as COPD is strongly influenced by environmental exposures, mainly cigarette smoke [28–30]. To determine if protease inhibitor (PI) type M, S or Z alleles influenced the *SERPINA1* distal ratio, we separately analyzed the distal ratio in individuals with MM, MS, MZ, and ZZ genotypes based on whole genome sequencing in our LTRC cohort data. This dataset includes genotyped MM individuals with no disease (n = 210) and with COPD (n = 284). Rare genotypes have proportionally lower representation in our data (MS: n = 14 and n = 20), (MZ: n = 8 and n = 16) and (ZZ: n = 1 and n = 7). As can be seen in Fig 2E, we observe a higher distal ratio amongst all genotypes with COPD (red box plots, Fig 2E). The distal ratio is significantly higher for individuals with COPD and an MS (p = 0.005) or MZ (p = 0.013) genotype compared to individuals with COPD and an MM genotype (Fig 2E). In the LTRC data set there were only n = 7 ZZ homozygous individuals, limiting the statistical power of this analysis (p = 0.15), but it is clear from Fig 2E that most ZZ individuals with COPD also have a higher distal. This indicates that COPD likely exacerbates use of the longer 3′ UTR. We further analyzed the distal ratio as a function of the Global Initiative for Chronic Obstructive Lung Disease (GOLD) spirometric stage [31] which shows a statistically significant

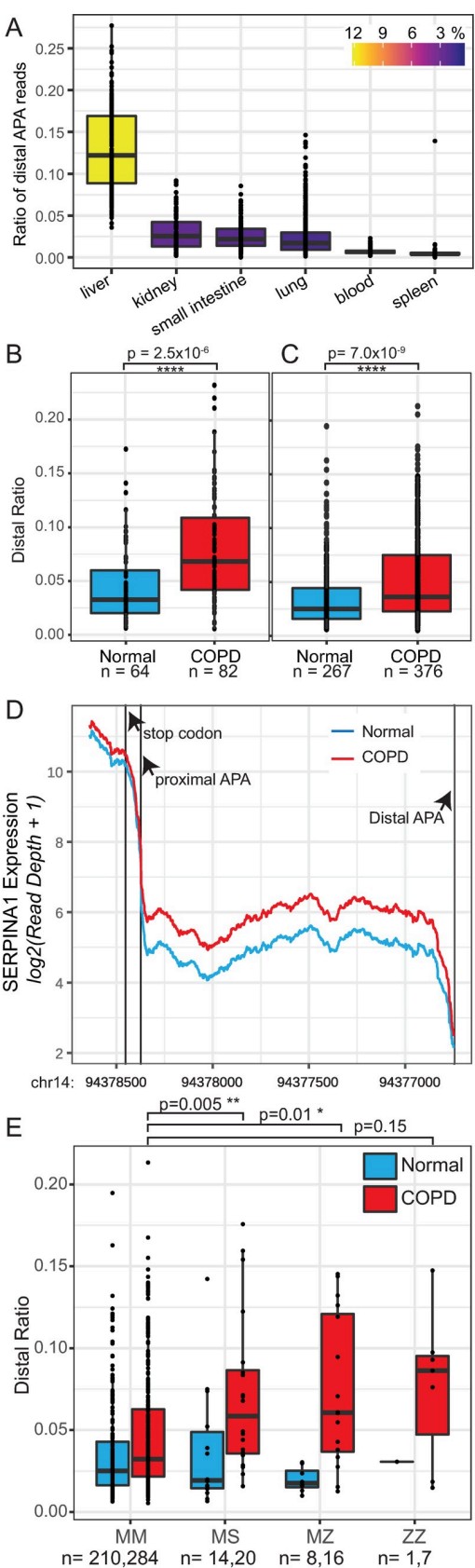

**Fig 2. Distal ratio across tissues and in COPD.** The distal ratio is measured as the relative depth of *SERPINA1* 3′ UTR distal vs. proximal reads in RNA-seq data. A) Distal ratio measured in the six tissues expressing *SERPINA1* mRNA above 1000 TPM in Gtex RNA-seq consortium data [10,11]. We observe the highest distal ratio in liver, but important variation in the lung. B) The distal ratio measured in a publicly available [26] short-read lung tissue RNA-seq data set from n = 82 (red) COPD subjects and n = 64 (blue) controls. C) Distal ratio measured from lung tissue RNA-seq data in the Lung Tissue Research Consortium for n = 376 COPD (red) subjects and n = 267 (blue) normal participants. In both studies we observe a significant increase of the distal ratio in COPD subjects. D) The characteristic drop at the proximal APA site indicative of alternative polyadenylation for mean COPD (red) and Normal (blue) lung tissue for n = 376 COPD subjects and n = 267 normal patients. E) Distal ratio analysis of LTRC subjects broken down by *SERPINA1* M (normal), S (mild disease) and Z (severe disease) alleles showing that the S and Z alleles exacerbate use of the distal alternative poly-adenylation site in the lungs of individuals with COPD.

increase in the distal ratio correlated with GOLD stage (S2 Fig). Together these data indicate that the distal ratio in lungs increases in individuals with COPD and increases with higher disease spirometric severity, raising the question of what the post-transcriptional functional effects of this lengthening are on A1AT expression.

## Role of long and short 3′ UTRs in *SERPINA1* translation and stability

To understand the functional consequences of APA site usage in the *SERPINA1* mRNA 3′ UTR we performed a series of assays using nanoluciferase reporter constructs in both HepG2 (liver hepatocyte) and A549 (adenocarcinomic human alveolar basal epithelial) cell lines. In these experiments we aim to measure the amount of protein produced for the short (proximal APA site used) and long (distal APA site used) isoforms (Fig 3A). In our construct design, the upstream APA site (proximal) is mutated (indicated by an X in Fig 3A) so that only the correct 3′UTR isoform can be expressed. Both constructs have an additional strong SV40 polyadenylation signal to ensure that the transcripts end as expected. Using 3′ end specific sequencing we confirmed that we can accurately detect 3′ ends (S3A and S3B Fig) and that short and long 3′UTR constructs have the expected sequence (S3C and S3D Fig). As can be seen in Fig 3B the proximal (or short) isoform yields 50-fold more luminescence relative to the distal (long) isoform. This is the case both in HepG2 and A549 cell lines.

To distinguish between changes in RNA stability and translational efficiency, we also measured the relative stability of the long and short isoforms in A549 cell lines. We observed a decline over time in both the long and short *SERPINA1* constructs (Fig 3C), consistent with high ΔCT values at 24 hours (S4A Fig). This finding indicates that the relative stability of the long and short isoforms (distal and proximal, respectively) is similar (Fig 3C). Longer 3′ UTRs are thought to destabilize mRNAs as they include more micro-RNA and RNA binding protein sites which affect mRNA stability and translation efficiency [12,32]. However, for *SERPINA1*, we do not detect significant stability differences between the short and long isoforms at early or late timepoints (Figs 3C and S4A). We find that steady state levels of the long isoform are lower than steady state levels of the short isoform (S4B Fig). However, this difference does not account for the large difference in translation between the short and long isoforms (Fig 3B). Thus, these experiments support the long isoform primarily inhibiting translation of the protein product.

## Mechanism of translation suppression by long SERPINA1 3′ UTR

To identify regions of the long *SERPINA1* 3′ UTR that control translation we designed a series of six deletion constructs based on the long 3′UTR construct that uses only a distal APA site (Figs 3D and S3). We measured the relative luminescence of these constructs in A549 cells and observed that in general shorter constructs resulted in higher protein production. However, the relationship between construct length and expression is not linear, rather each deleted

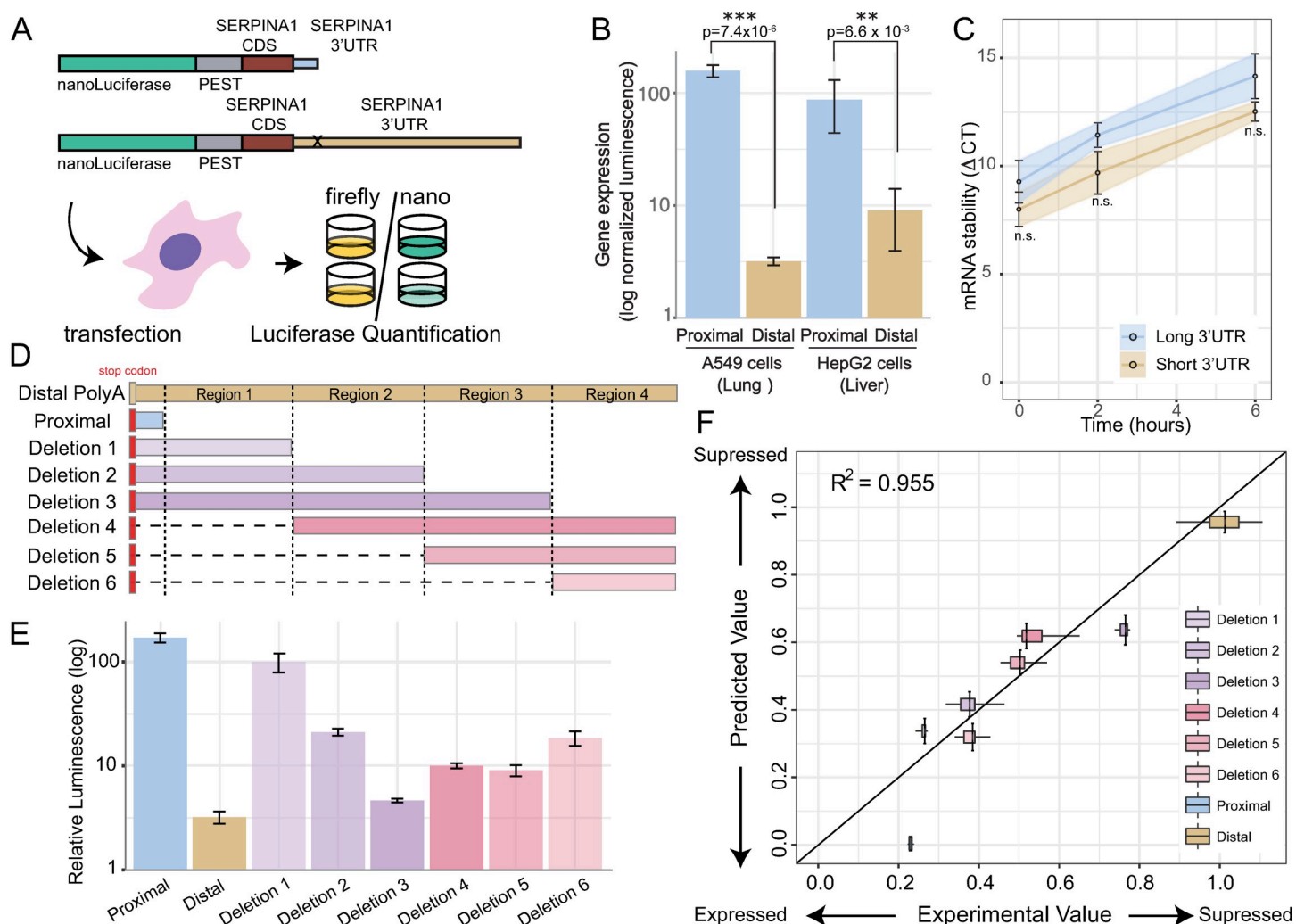

**Fig 3. Luciferase reporter and mRNA stability assays to measure the effect of long vs. short 3′ UTRs.** To measure effect on translation efficiency and mRNA stability of the long and short 3′ UTR sequences we performed a series of luciferase reporter assays. A) Schematic of the luciferase reporter assay, combining a nanoluciferase reporter, PEST domain and the *SERPINA1* exon 7 coding sequence upstream of the long and short 3′ UTRs. The proximal APA site is mutated to inhibit use of this site (indicated with an x on the long construct). These constructs are co-transfected in HepG2 and A549 cell lines with a control firefly reporter and the ratio of nanoluciferase protein to firefly protein measured. B) Log normalized luminescence, which indicates gene expression, measured for short (blue) and long (tan) *SERPINA1* isoforms. The expression is significantly higher for the short isoform by close to two orders of magnitude in both lung derived A540 cells and liver derived HepG2 cells. C) Pulse chase experiments in A549 cells using ethylene uridine (EU) and click-it chemistry for labeling with biotin-azide to measure relative mRNA stability by qRT-PCR [63]. We confirmed that GAPDH was consistently stable over the time period and similar declines in both the long and short *SERPINA1* RNAs, indicating similar stability for both long and short 3′ UTRs. D) Deletion construct design to identify regions controlling gene expression in *SERPINA1* 3′ UTR. Six constructs were designed to selectively delete regions 1–4. E) Relative luminescence which indicates expression for the six deletion constructs compared to short (blue) and long (tan) *SERPINA1* 3′ UTRs. F) Predicted vs. measured expression of deletion constructs using regression model described by Eq 1. This model yields the translation coefficients of the four regions reported in Table 1.

**Table 1. Translation Coefficient of SERPINA1 3′ UTR Regions.**

| Region | Transcript Coordinates | Genomic Coordinates | Translation Coefficient | eCLIP RBPs |
|---|---|---|---|---|
| Proximal | NM_000295:1519–1597 | chr14:94378451–94378373 | - | PCBP1, PCBP2, SND1, LARP4 |
| Region 1 | NM_000295:1619–2016 | chr14:94377952–94378350 | 0.34±0.04 | AKAP1, PCBP2, ILF3 |
| Region 2 | NM_000295:2017–2416 | chr14:94377953–94377554 | 0.08±0.05 | AKAP1, PCBP2 |
| Region 3 | NM_000295:2417–2816 | chr14:94377555–94377156 | 0.22±0.05 | AKAP1, PCBP2, ILF3, QKI |
| Region 4 | NM_000295:2817–3220 | chr14:94377157–94376747 | 0.32±0.04 | QKI |

region appears to affect translation to a different degree (Fig 3E). To understand the role of each region in controlling translation we developed a model to predict the relative contribution to translation efficiency of each specific region (Eq 1). When we fit this model to our data (Fig 3F) we obtained high correlation between predicted and experimental values ($R^2$ = 0.96). Furthermore, an analysis of the relative weight of each region's translation (Table 1) reveals that no single region uniquely controls translation efficiency. Nonetheless, regions 1, 3 and 4 together account for over 90% of the variation in translation efficiency between the long and short 3′ UTRs. Together these experiments suggest that the primary function of the long (distal) 3′ UTR in *SERPINA1* is to inhibit translation, and that regions 1, 3, and 4 contribute the most to the translation repression. To dissect the mechanisms controlling translation repression and distal vs. proximal APA site usage we now investigate the effects of *trans* factors on the *SERPINA1* 3′ UTR.

## Role of RNA Binding Proteins and structure in controlling translation efficiency

The fact that the long isoform of the 3′ UTR does not destabilize the message (Fig 3C) but decreases translation efficiency by several orders of magnitude is uncommon, but not unprecedented [33]. To reveal a mechanism that can account for this we begin our investigation by mining eCLIP data (enhanced CrossLinking and ImmunoPrecipitation) from recent ENCODE (Encyclopedia of DNA Elements) experiments on RNA binding proteins carried out in HepG2 cell lines [34–36]. In Fig 4A, we show high-confidence experimental eCLIP binding sites on the long SERPINA1 3′ UTR isoform. Several interesting features emerge from this analysis. First, multiple proteins bind near the proximal APA site, but further downstream fewer binding sites are present. The most distal binding site identified by eCLIP is for the RNA binding protein QKI (Quaking homolog) which spans regions 3 and 4. No other RBPs were identified by eCLIP binding downstream of this site. As a reminder, Deletion 1 (which removes Regions 2, 3 and 4) restores the translation efficiency of *SERPINA1* 3′ UTR to near proximal levels (Fig 3E). Also, deletions 5 and 6 (which maintain regions 3 and 4) both suppress translation to near full-length levels (Fig 3E). While other RNA binding proteins may have a role in regulating *SERPINA1* mRNA or A1AT expression [37–39] (S1 Table), QKI's distal binding position in Regions 3 and 4 in the 3'UTR is unique.

Our previous work on the *SERPINA1* 5' UTR established that RNA structure plays an important role in controlling translation efficiency of the A1AT protein [4]. To determine if RNA structure could play a role in controlling translation by mediating QKI binding, we performed SHAPE-MaP (Selective 2' Hydroxyl Acylation by Primer Extension and Mutational Profiling) to determine the secondary structure of *SERPINA1* 3′ UTR [40–43]. The SHAPE data is shown in Fig 4B, where red bars indicate high, yellow intermediate, and black low nucleotide reactivity. High SHAPE reactivities indicate the nucleotides are flexible and therefore likely unpaired [4,41,42]. From the SHAPE data we compute the Shannon Entropy of the RNA, which measures the degree of structuredness (Fig 4C) [42–45], as well as the structure indicated as an arc-plot in Fig 4D. We observe low Shannon Entropy near the distal APA site and in the short isoform, suggesting a high level of RNA structuredness in the short isoform but high entropy (low structuredness) in Regions 2, 3 and 4. Interestingly, the QKI site has low Shannon Entropy indicative of higher structure near the binding site. Previously published analysis of QKI *in vitro* RNA binding specificity suggests that the protein binds to the consensus motif 5'-NACUAAY-N(1,20)-UAAY-3' [46]. The motif is present in the *SERPINA1* 3′ UTR within the eCLIP QKI binding site, and multiple nucleotides have high SHAPE reactivity in the motif suggesting it will be accessible for binding (Fig 4E).

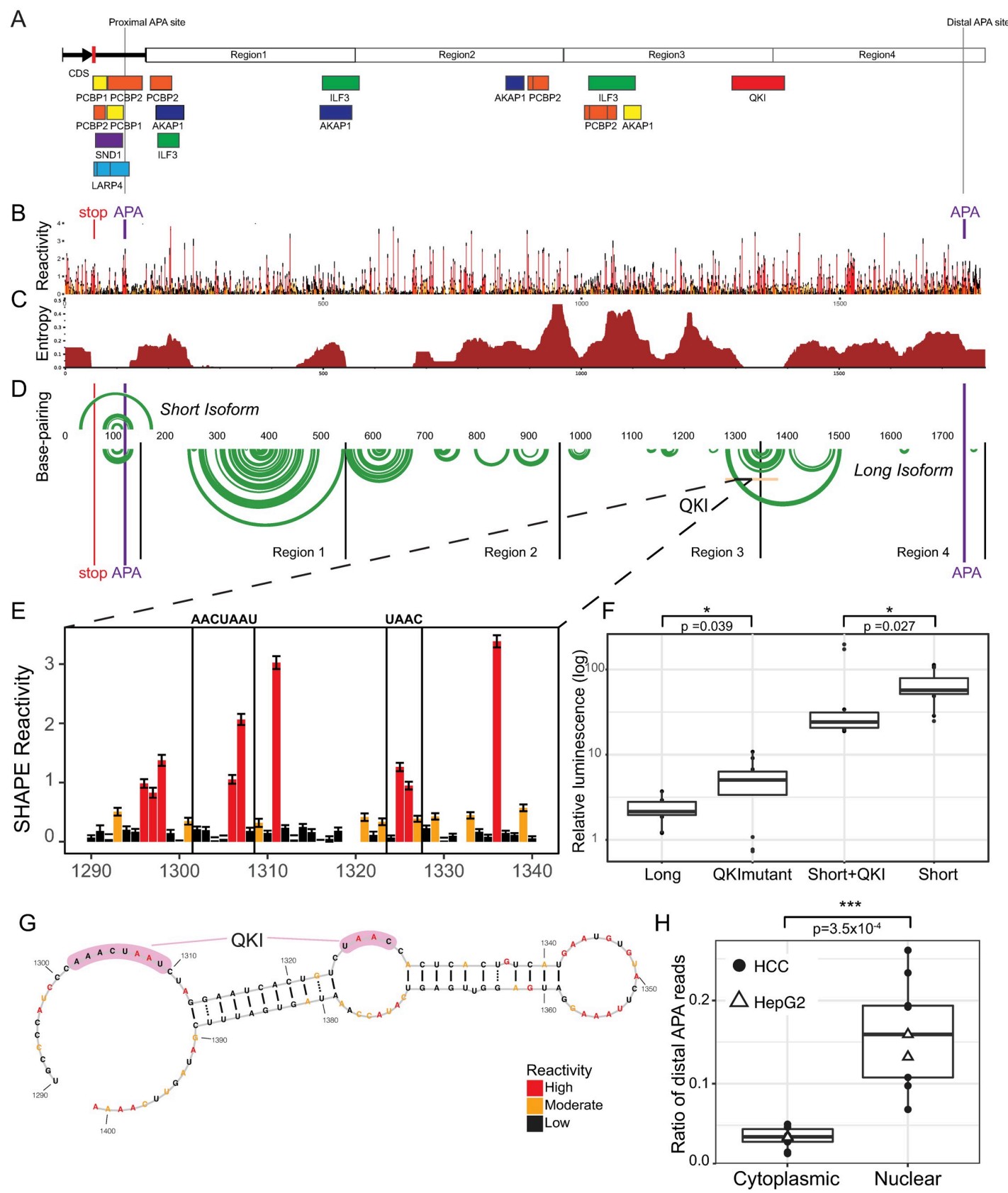

**Fig 4. Role of RNA binding proteins and structure in modulating translation efficiency.** Mapping of eCLIP (enhanced CrossLinking and ImmunoPrecipitation) from recent ENCODE (Encyclopedia of DNA Elements) experiments on RNA binding proteins carried out in HepG2 cell lines [34–36] onto the long 3′ UTR isoform of *SERPINA1* mRNA. Each rectangle indicates a protein binding site and sites are colored by RBP. B) SHAPE (Selective 2' Hydroxyl Acylation by Primer Extension) structure probing long *SERPINA1* 3′ UTR. SHAPE data identifies flexible (unpaired) nucleotides in the RNA structure revealing regions of high accessibility for protein binding. Red indicates highly reactive nucleotides, yellow intermediate and black low. C) Using SHAPE data, we compute the entropy of the *SERPINA1* mRNA structure. Low entropy regions adopt single, well-defined structures, while high entropy regions are more disordered [64,65]. D) SHAPE derived secondary structure model for short (top) and long *SERPINA1* 3′ UTR indicated as an arc diagram. Only highly probable base-pairs are shown (green). As can be seen QKI (Quaking Homolog) binds to a low-entropy region in the 3′ UTR with extensive local base-pairing. E) SHAPE reactivity for the nucleotides for the eCLIP QKI binding site including the putative binding motif of the RNA binding protein 5′-NACUAAY-N(1,20)-UAAY-3′ [46]. F) We found the QKI binding site to impact gene expression from the *SERPINA1* 3'UTR using our nanoluciferase assay. Mutating the QKI binding site increases expression from the long 3'UTR isoform (QKImutant) while adding the QKI binding site and flanking nucleotides decreases expression of the short 3'UTR isoform (Short+QKI) in A549 cells. G) Secondary structure model derived from experimental SHAPE data for QKI binding region, both regions of the motif are accessible for binding. H) QKI is known to retain mRNAs in the nucleus. We measured the *SERPINA1* distal ratio in Hepatocellular carcinoma (•) and HepG2 (Δ) nuclear and cytoplasmic RNA-seq fractions and find the highest ratio in the nucleus. This suggests QKI binds and retains the long *SERPINA1* isoform in the nucleus thereby inhibiting translation.

The eCLIP data, the presence of the QKI consensus motif in the *SERPINA1* 3′ UTR, SHAPE structural analysis, and our luciferase data suggest QKI is likely playing a central role in translation repression of A1AT expression. Using our nanoluciferase translation assay, we therefore mutated the bipartite QKI motif in the long isoform, which significantly restores translational efficiency (Fig 4F, QKImutant). Addition of the QKI bipartite motif to the short isoform (Short+QKI) significantly represses translation (Fig 4F). These data therefore further support that QKI is directly affecting translation of *SERPINA1* 3′ UTR.

Finally, we modeled the RNA secondary structure of the *SERPINA1* 3′ UTR (Fig 4G) using the SHAPE data as a pseudo-free energy as previously carried out in [42,47,48]. This structure has potential ramifications as a modulator of translation repression and would also be a logical RNA target for small molecule or anti-sense therapeutic development [49]. We observed significant base-pairing in Region 1 and several hairpins spanning Regions 3 and 4 (indicated as green arcs, Fig 4D), which are consistent with low Shannon Entropy observed for the QKI binding site. Our structural model shows the proximal APA site is accessible (S1B Fig).

One known function of QKI is to retain transcripts in the nucleus, which has the effect of repressing translation [50]. Thus, a majority of the *SERPINA1* long 3′ UTR should be retained in the nucleus if this is the mechanism of translation suppression. When we analyze the distal ratio from cytoplasmic and nuclear poly-A selected fractions of HepG2 and HCC (Hepatocellular carcinoma) RNA-seq data [51,52] we observe a much higher proportion of distal reads in the nuclear fraction (Fig 4H). Based on the increased ratio of distal reads in the nucleus, we posit that one mechanism by which QKI could be altering translational efficiency is by retaining *SERPINA1* mRNA in the nucleus. This would effectively inhibit translation since mRNAs must be exported to the cytoplasm to be translated by the ribosome. Additional mechanisms may be involved in translational inhibition in other regions of the 3'UTR that we did not detect since the ENCODE RNA binding protein data is not a comprehensive atlas of all known RNA binding proteins. Nonetheless, our findings demonstrate the value of integrating large scale genomic data sets in investigating post-transcriptional regulation.

## Differential expression of RBPs in COPD and the distal ratio

Having analyzed eCLIP RNA binding protein data, we were interested in identifying RBPs that alter the distal ratio of the *SERPINA1* 3′ UTR in COPD that may not have been detected by RNA binding protein crosslinking. We therefore measured differential expression in the LTRC data for the 224 RBPs for which shRNA ENCODE knock-down data is available [34–36]. For these 224 RBPs we also measured the distal ratio upon shRNA knock-down and plot this on the y-axis of Fig 5A. The plot is centered on the control, empty vector shRNA distal ratio, indicated by a horizontal line. On the x-axis we plot the $\log_2$ differential expression fold

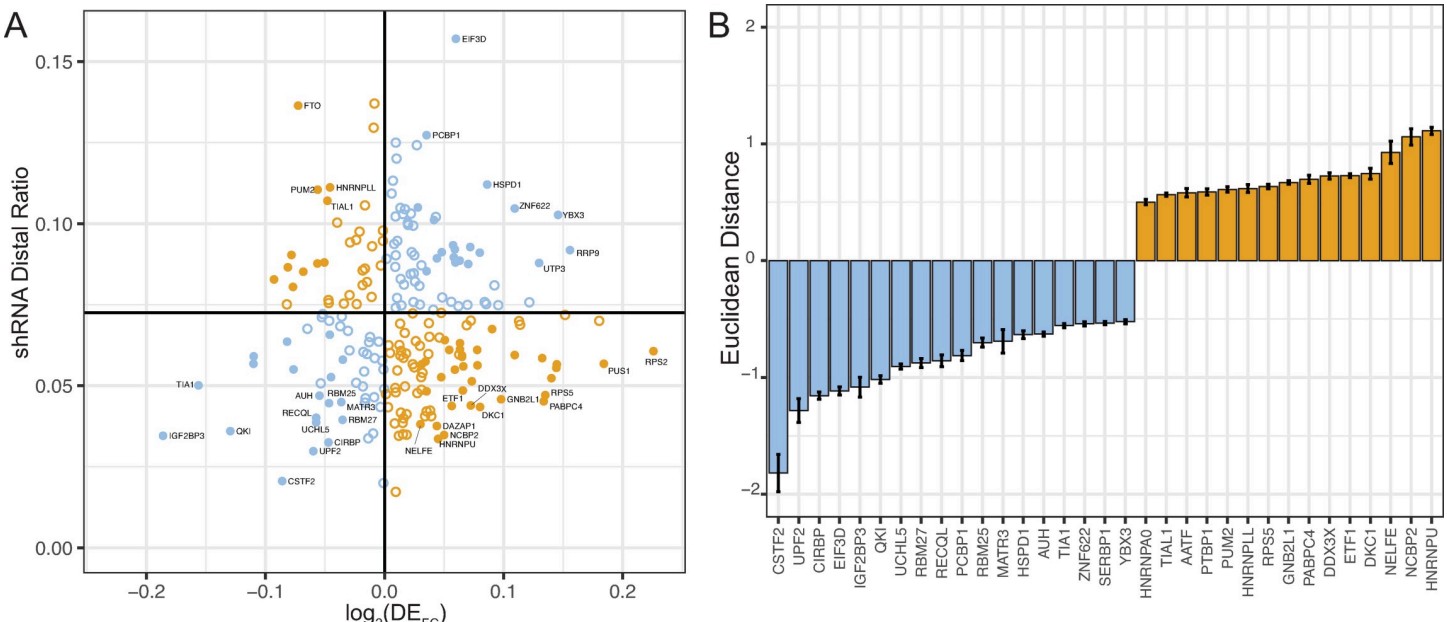

**Fig 5. Role of RNA binding proteins and relative single-cell populations in modulating *SERPINA1* APA.** A) Two-dimensional volcano plot of endogenous shRNA distal ratio in *SERPINA1* (y-axis) as a function of $\log_2$ differential expression fold change ($\log_2(DE_{FC})$) in LTRC primary lung tissue from 376 COPD cases and 267 controls for the 224 corresponding RNA binding proteins. The horizontal line represents the mean distal ratio in the corresponding empty vector shRNA controls, while the vertical line is centered on zero. Blue points (bottom right quadrant and top left quadrant) will decrease the distal ratio in COPD, while orange points (top left and bottom right quadrant) will increase the distal ratio. Filled circles indicate RNA binding proteins for which the distal ratio changes from shRNA control, and $\log_2(DE_{FC})$ are both significant in both data sets with $p_{adj} < 0.05$. B.) $\log_2$ computed Euclidean distance from center of two-dimensional volcano plot for significant ($p_{adj} < 0.05$) RNA binding proteins affecting COPD. Again, those RNA binding proteins expected to lower the distal ratio are colored blue and negative, while those expected to increase the distal ratio are orange and positive.

change $\log_2(DE_{FC})$ for these 224 RBPs when comparing the 376 COPD cases and 267 controls in the LTRC primary lung tissue data. In Fig 5A, the further the RBP is from the center of this two-dimensional volcano plot the more likely it modulates the distal ratio in COPD lungs.

When we measure the distal ratio change in shRNA knockdown experiments (y-axis, Fig 5A), we are reporting the change in distal ratio upon down-regulation of that RBP. These same RBPs may be up, down, or not regulated in the lung tissue of COPD vs. control individuals. As we consider the representation developed in Fig 5A, there are two types of proteins: 1.) RBPs that increase the distal ratio (e.g. because their distal ratio is above the control ratio and $\log_2(DE_{FC}) < 0$ in COPD), or 2.) RBPs that decrease the distal ratio in COPD (e.g. because their measured distal ratio after shRNA is below that of the control and $\log_2(DE_{FC}) < 0$). Thus, RBPs in the top left and bottom right corners of Fig 5E contribute to increasing the distal ratio in COPD (orange), and those in the bottom left and top right corners contribute to decreasing the distal ratio (blue). In Fig 5A, filled circles meet the $p_{adj} < 0.05$ for both distal ratio difference from control and $DE_{FC}$. Numerical data used to create Fig 5A is also provided in S2 Table. We notice that QKI is in the bottom left quadrant as it is significantly down regulated in COPD ($\log_2(DE_{FC}) = -0.13$, $p_{adj} = 5.4 \times 10^{-16}$) and significantly decreases the distal ratio in shRNA knockdown experiments (distal ratio = 0.036, $p = 1.54 \times 10^{-32}$). QKI decreased expression in COPD therefore decreases the long isoform of the *SERPINA1* mRNA, which would lead to higher translation efficiency of the A1AT protein.

One clear result in Fig 5A is that no single protein can account for the increase in usage of the distal 3′ UTR site we report in Fig 2. To quantify the effect of each protein we compute the $\log_2$ Euclidean distance from the center of the graph (Fig 5B) which quantifies the potential

effect of each RNA binding protein on the distal ratio in COPD. A higher distance indicates a higher effect in COPD of the RBP on the distal ratio. We plot this distance as negative if the RBP lowers the distal ratio in COPD (blue) and positive (orange) if it increases. If we consider all RBPs where the absolute value of the $\log_2$ Euclidean distance $> 1$ and $p_{adj} < 0.05$ for shRNA distal ratio change and $\log_2(DE_{FC})$ then we identify 6 RBPs that decrease the distal ratio and only 2 that increase it (Fig 5B blue and orange, respectively, S2 Table). CSTF2 (cleavage stimulation factor 2), an important modulator of the cleavage stimulation factor complex which regulates APA is in the bottom left-hand corner and has the most negative Euclidean distance in Fig 2B, such that its differential expression in COPD will also contribute to less distal *SERPINA1* mRNA expression.

This analysis reveals that in COPD lung tissue, RNA binding protein expression levels are adjusted in a way that favors the short *SERPINA1* isoform, thereby resulting in higher A1AT translation efficiency. This is contrary to what we observed in Fig 2, where the distal ratio is increased in COPD individuals. Furthermore, *SERPINA1* mRNA is not differentially expressed in COPD LTRC data ($\log_2(FC) = 0.07$, $p_{adj} = 0.16$), and that these changes only affect the ratio of long to short isoform, and therefore the translation of the A1AT protein in the lung. An important caveat in interpreting the data in Fig 5 is that it is not a comprehensive analysis of all known RNA binding proteins, since we are limited on the y-axis by existing ENCODE shRNA experiments. Furthermore, the shRNA data were collected in HepG2 cells, whereas our differential expression analysis was measured in primary lung tissue. There are therefore likely additional RBPs controlling the distal ratio in the lung that this analysis cannot reveal. It is possible that a different molecular mechanism this analysis cannot reveal explains the observed lengthening in COPD lungs. However, our analysis in Fig 5A and 5B of expression changes in the LTRC lung cohort reveal that in COPD lungs, multiple RNA binding proteins (including QKI and CSTF2) have altered expression that, in the context of corresponding shRNA knock-down experiments, will lead to a decreased distal ratio. This may act as a molecular buffer, effectively keeping the distal ratio from increasing more.

We also analyzed single-cell RNA-seq data to investigate if another mechanism could explain the observed higher distal ratio observed in Fig 2B and 2C. Single-cell RNA-seq from five healthy human liver biopsies previously identified 20 clusters of cells, including six different hepatocyte clusters, shown in S5A Fig [53]. *SERPINA1* mRNA is most predominantly expressed (green) in the six hepatocyte clusters (cluster numbers 1, 3, 5, 6, 24 and 15) (S5C Fig). In comparison lung UMAP (Uniform Manifold Approximation and Projection) single cell RNA-seq analysis tissue from five healthy donors previously revealed 25 cell type clusters (S5B Fig) [54]. In these data *SERPINA1* mRNA is predominantly expressed in Alveolar Type 1 and 2 cells as well as Macrophages (S5D Fig).

Read coverage in single-cell RNA-seq data is significantly sparser and has higher variability than bulk RNA-seq data. We were able to visualize the distribution of distal ratios for each cell type in clusters with sufficient cells expressing *SERPINA1* mRNA. For the liver data these include three hepatocyte clusters (1, 3, and 6) as well as Macrophages, αβ-Tcells and Plasma cells (S5E Fig). In the lung, these are Alveolar Type 1 and 2, Dendritic cells and Macrophages (S5F Fig). We observe that the distal ratio distributions are different in each cell type, but these differences are not statistically significant in these data, mostly due to very large variations observed. Recent single-cell analysis of lung tissue in COPD individuals observed important changes in AT2 cell populations in COPD individuals [55], however, the methods used for cell dissociation do cause substantial losses in AT1 cells. This makes it difficult to quantitatively relate these results with our observations in the LTRC data. Nonetheless, any changes in relative cell-type abundance in COPD lungs will affect the observed distal ratio in bulk-RNA seq data, and we hypothesize this is the mechanism that causes the observed increase in distal ratio

observed in the LTRC COPD data (Fig 2C). Although we cannot quantitatively establish that RBPs or changes in cell type population alone are responsible for the observed difference in the distal ratio of *SERPINA1* mRNA in COPD lungs (Fig 2B, 2C and 2D), our data reveal the complex network of factors that ultimately control A1AT protein production at the molecular and cellular level in the lung.

## Discussion

A1AT protein expression is key to both liver and lung disease, and deficiency of the protein can lead to panlobular emphysema [7–9]. In the liver, Z-allele individuals often accumulate polymers of mutant A1AT leading to liver cirrhosis [2,56]. Although the *SERPINA1* mRNA is expressed at the highest level in the liver, specifically in hepatocytes (S5C Fig), we show here it is also expressed in primary lung tissue, in Alveolar Type 1, 2 and macrophage cell types. The prevailing understanding of A1AT function is that it is secreted from the liver and carries out its functions in the lung [7,8]. Our findings agree with this, but also reveal a potentially important role for *SERPINA1* mRNA in the lung. The unique transcript structure of the A1AT mRNA, with 11 transcript isoforms differing only in their 5' UTR, is indicative of a post-transcriptional regulatory mechanism encoded in the non-protein-coding regions of this mRNA [3,4,6]. We report here the characterization of an alternative poly-adenylation event in the *SERPINA* 3′ UTR that controls A1AT expression.

Our measurements of both short and long-read RNA-sequencing, combined with 3′ end-sequencing confirms that the *SERPINA1* 3′ UTR has a long (distal) and short (proximal) isoform (Fig 1). Use of the distal site is significantly increased in lung tissue from COPD subjects (Fig 2) and this yields transcripts with a 50-fold lower translation efficiency (Fig 3). Furthermore, the heterozygous MS and MZ individuals and ZZ individuals with COPD all present a higher distal ratio compared with MM individuals with COPD. Given that a higher distal ratio results in lower A1AT translation, our work indicates that alternative polyadenylation, and in particular the use of the distal site in these individuals, likely exacerbates A1AT protein deficiency in these individual's lungs. Thus, the post-transcriptional response in the lung further prevents the production of A1AT protein, presumably where A1AT produced in the liver and transmitted through the plasma does not have access.

Our analysis of *SERPINA1* 3'UTR translation, structure, sequence, and interactions with RNA binding proteins (Fig 4) indicate that QKI suppresses A1AT protein expression from *SERPINA1* isoforms with long 3'UTRs. These data agree with shRNA knockdown experiments where we observe a lower distal ratio in the absence of QKI. Nuclear retention is a known function of QKI in the brain and may be part of the translational inhibition of *SERPINA1* translation in liver and lung tissue [50,57]. Interestingly, QKI is downregulated in COPD lungs, which suggests that at a molecular level, individuals with COPD retain the long isoform less in their lungs. Our structural results also suggest that targeting the RNA structure near the QKI binding site in the *SERPINA1* 3′ UTR would likely increase translation of the A1AT protein, providing a potentially novel site for RNA therapeutic development that could be particularly advantageous to MS, MZ and ZZ individuals with COPD [49].

Interestingly, mice have six *SERPINA1* paralogs (Serpina1a-f), which have been edited out to generate a model of A1AT deficiency and emphysema [58]. While all these paralogs share a conserved proximal polyadenylation site with the human *SERPINA1* (S6A Fig), none of the murine paralogs contain a QKI binding site anywhere in the potential 3'UTR (S6B Fig). Only *Serpina1c*, *d* and *f* have potential distal sites (S6C Fig). *Serpina1c* and *Serpina1d*, as most of the other murine paralogs, have their highest expression in liver, while *Serpina1f* is expressed in the kidney (S6D and S6E Fig). However, there is no evidence for significant expression of the

distal region in any of the murine *Serpina1* paralogs in two different RNA-Seq datasets (median distal ratio < 0.001, S6F and S6G Fig). Post-transcriptional regulation of *SERPINA1* has therefore diverged between humans and mice, and testing of *SERPINA1* therapeutics aimed at modulating RNA regulation will require development of a different animal model system.

Our data reveal a more fundamental aspect of post-transcriptional response to COPD in lung tissue. We found that CSTF2 is significantly down regulated in COPD lungs ($\log_2(DE_{FC})$ = -0.09, $p_{adj}$ = 9.5x10$^{-5}$). This RNA binding protein is a global regulator of transcription termination and plays a central role in controlling alternative poly-adenylation [59–61]. For the *SERPINA1* mRNA, the result is a preferential use of the proximal site, which in turn results in higher A1AT expression. In fact, most changes in RNA binding protein expression in COPD lungs (Fig 5B) have the effect of favoring the proximal alternative polyadenylation site. Thus, the molecular response in the lung at the level of RNA binding proteins favors the short 3′ UTR of the *SERPINA1* mRNA, increasing the translation and expression of the A1AT protein.

The skew toward RBPs that decrease the amount of *SERPINA1* long 3′UTR isoforms in individuals with COPD (Fig 5A and 5B) contradicts our observation of the 2.1-fold higher distal ratio (greater use of the distal site) in COPD lungs (Fig 2). However, our analysis of single-cell RNA-seq data from primary lung and tissue data reveal that *SERPINA1* mRNA is expressed in multiple different cell types. Although single-cell RNA-seq data is still too sparse allow us to accurately measure the distal ratio for each cell type cluster, we observe variation in the distal ratio between individual cells. Thus, one possible explanation for the increased distal ratio observed in COPD lung tissues is a change in cellular composition. This change may also be driven by differences in the inflammatory state of the lung. If this is indeed the case, then our finding that RNA binding protein expression changes in COPD lungs to increase the shorter isoform indicates that molecular expression effectively balances changes in cellular composition and/or inflammation that increase 3′ UTR length. This may also explain why we observe only a 2.1-fold increase in distal ratio in COPD, as it is effectively buffered by compensating changes in RNA binding expression. Finally, the ENCODE RNA binding protein eCLIP and shRNA data is very powerful but is not a comprehensive atlas of all known RNA binding proteins. As such, it is possible that a yet to be profiled RNA binding protein lengthens the 3' UTR that our analysis could not reveal.

Although statistically significant, the 2.1-fold increase in distal site usage in COPD lungs remains modest (Fig 2) and would have at most a 15% decrease on the overall translation and expression of A1AT protein. Given the 50-fold decrease in translation efficiency of the long isoform 3′ UTR isoform, it is however essential to buffer against a high distal ratio in all tissues requiring protein expression. Our data reveal that RNA binding proteins, QKI and CSTF2 play a central role in this regulation and are accordingly differentially regulated in the lungs of COPD individuals. Importantly, our findings are also consistent with one known mechanism of translational control by QKI, which is nuclear retention. Independent of the exact mechanism, our data suggest that inhibition of the QKI/*SERPINA1* 3'UTR interaction offers a potential therapeutic strategy for increasing A1AT expression in the lungs of individuals with COPD.

## Methods

### Cell culture

Liver derived hepatocellular carcinoma (HepG2) cells and lung derived epithelial carcinoma (A549) cells were obtained from ATCC through UNC Tissue Culture Facility. HepG2 cells were cultured in Eagle's Minimum Essential Media supplemented with 10% fetal bovine

serum. A549 cells were cultured in RPMI 1640 media supplemented with 10% fetal bovine serum. Cells were split regularly using Tryple (Fisher). All cells were grown in 5% $CO_2$ and 37°C.

### 3′ sequencing

Total RNA was extracted from HepG2 cells using trizol (ThermoFisher) followed by RNA column purification (Purelink Invitrogen) and on-column DNA digestion (Purelink, Invitrogen). RNA was used to create 3′ end specific libraries with the Quant Seq kit (Lexogen). The same sample of HepG2 RNA was depleted of ribosomal RNA (Ribominus Eukaryotic v2, Fisher) and used to generate standard Illumina short-read libraries using the Nextera Flex kit (Illumina). Both libraries were sequenced (Illumina MiSeq, 300 kit) and the fastq sequence files used in downstream analysis. The standardized Bluebee Lexogen Quant Seq FWD protocol was used for trimming, read alignment, and quality control steps of the Quant Seq library. The standard HepG2 transcriptome sequencing was analyzed in the same manner as other fastq datasets (see—Calculation of *SERPINA1* distal ratio from sequencing).

### Calculation of SERPINA1 distal ratio from RNA sequencing

We followed the GTEx Consortium and TopMed guidelines for sequence alignment to the human genome (https://github.com/broadinstitute/gtex-pipeline/commits/master/TOPMed_RNAseq_pipeline.md). Briefly, we used STAR [66] to align fastq files to hg38 (containing no alternative, no decoy and no HLA chromosomes). After alignment, or with pre-aligned samples, we used samtools to determine the read depth at each nucleotide in the 3′UTR, summed up the depth within the distal and proximal specific regions and normalized by the gene region length. We calculated the ratio of distal 3′UTR reads by taking the length normalized read counts in the distal region and dividing by the sum of the length normalized distal and proximal reads. The same approach was used on individual clusters for single-cell RNA-seq analysis (S5 Fig).

### Construction of nanoluciferase SERPINA1 3′UTR reporter system

We gene synthesized the full 3′UTR sequence of SERPINA1 (Twist biosciences) with a mutated proximal polyA site (AUUAAA to AUCAAG) and 60 nucleotides of coding sequence. This coding region and full 3′UTR were cloned into pNL3.2.CMV (Promega) using sequence homology (NEB Builder HiFi). A short 3′UTR isoform was amplified from the gene synthesized construct using PCR with the same coding sequence portion. This short isoform was cloned into the same backbone with sequence homology. The nanoluciferase expressed from this construct contains a PEST domain to prevent protein accumulation. All additional deletion and mutation constructs were created from verified long 3'UTR plasmids. Primers for cloning are listed in S3 Table. The mutated polyA site was reverted back to wildtype for both the long and short 3′UTR constructs (NEB Q5 Site directed mutagenesis kit) for structural studies. All constructs were verified by Sanger sequencing and restriction digest.

### Luciferase experiments

For deletion series analysis, A549 cells were split into 96 well opaque, clear bottom plates (~150,000 cells/mL) and transfected the next day with 10 ng of control firefly luciferase and 10 ng of nanoluciferase constructs using standard quantities of Lipofectamine 3000 and P3000 in serum-free Optimem (Fisher). HepG2 cells were treated similarly, but 50 ng of firefly and nanoluciferase constructs were used. Cells were transfected in triplicate. After 30 hours plates

were used to measure luminescence using the Promega Nano-Glo Dual-Luciferase Reporter Assay System. Luminescence was quantified at 480 nm on the BMG Labtech Clariostar plate reader. Four biological replicates from separate days were combined for analysis. R was used to normalize samples (0 to 1) and we created a linear model with no y-intercept. Coefficients for each region are listed in Table 1. For QKI mutation analysis, A549 cells were split into 12 well plates and transfected the next day with 0.5 ug of control firefly luciferase and 0.5 ug of nanoluciferase constructs using standard quantities of Lipofectamine 3000 and P3000 in serum-free Optimem (Fisher). Cells were transfected in triplicate. After 48 hours plates were used to measure luminescence using the Promega Nano-Glo Dual-Luciferase Reporter Assay. Luminescence was quantified at 480 nm on the SpectraMax iD5 plate reader. Three replicates from separate days were combined for analysis.

$$Model = t_1 Region1 + t_2 Region2 + t_3 Region3 + t_4 Region4 \tag{1}$$

## RNA stability quantification

We transfected A549 cells (same conditions as Luciferase assays) with equimolar amounts of either the long or short 3′UTR nanoluciferase reporter and performed a pulse-chase experiment by incubating transfected cells with ethylene uridine (EU) and harvesting either immediately, two hours or six hours later (Click-iT Nascent RNA Capture Kit, Invitrogen). Incorporation of EU into RNA does not alter cellular biology and allows for click-it chemistry and labeling with biotin-azide [63]. RNA was extracted using Trizol (Invitrogen) chloroform extraction and Phaselock Tubes (Invitrogen). RNA was purified using Purelink RNA Mini Kit (Invitrogen) and treated with DNAse I (Invitrogen) prior to being converted to cDNA using VILO reverse transcriptase according to manufacturer's instructions (Invitrogen). qPCR was carried out by using PowerUp Sybr Green Master Mix (Thermo Fisher Scientific) on cDNA isolated from 0, 2 and 6 hour time points and primers specific for nanoluciferase, the short or long isoform of *SERPINA1*. Primers are listed in S3 Table. The qPCR reactions were performed on an Applied Biosystems QuantStudio 6. We confirmed that GAPDH was consistently stable over the entire time period and then determined SERPINA1 3′UTR RNA stability by normalizing expression values of triplicate samples to GAPDH ($\Delta CT = CT_{SERPINA1} - CT_{GAPDH}$).

## RNA structure experiments

A T7 promoter was added to the wild-type nanoluciferase SERPINA1 3′UTR during PCR amplification. *In vitro* RNA was produced off the purified PCR product (NEB HiScribe T7 High yield RNA synthesis kit), treated with Turbo DNase, purified and incubated at 37°C for 10 min in folding buffer (final concentration, 200 mM Bicine, 100 mM NaCl and 10 mM MgCl2). SHAPE treatment was performed with an addition of DMSO (control, 10% final concentration) or 5NIA (25 mM in DMSO final concentration) for 5 min at 37°C. The RNA was purified (AmpureXP RNA beads). We generated sequencing libraries (Swift RNA library kit), bioanalyzed and quantified the libraries and then sequenced the samples (Illumina MiSeq 300 kit). Reactivity profiles were generated using SHAPEMapper [40]. Base-pairing probabilities were generated with SuperFold and RNA structure models with RNAstructure [67,68].

## RBP shRNA knockdown effects and eCLIP analysis of SERPINA1 3′UTR isoforms

The RBP eCLIP binding data was downloaded from the ENCODE data portal on Nov. 18, 2019. This download was restricted to assays in K562 and HepG2 cell lines and for RBPs with a

"Target Category" in ENCODE that matched "RNA binding protein" and mapped to GRCh38. This resulted in eCLIP data for 122 RBPs. All downloaded eCLIP sites were originally identified through the described ENCODE 3 pipeline, a stringent process that takes into account experimental reproducibility and fraction of reads in peaks (FRiP) over background [34]. We then mapped peaks using the narrowPeak bed files with "1, 2" listed in the biological replicates column in the downloaded ENCODE metadata file. We also downloaded sequencing files for RBP shRNA knockdown for control shRNA treatment and shRNAs against SERPINA1 RBPs in HepG2 cells from the ENCODE portal. All shRNA experiments had a knockdown efficiency of at least 50% compared to the scrambled control for protein and RNA. We aligned these fastq files and calculated the ratio of distal 3′UTR reads (see—Calculation of SERPINA1 distal ratio from sequencing).

### RBP differential expression analysis in LTRC data

COPD was defined as forced expiratory volume in one second ($FEV_1$) to forced vital capacity (FVC) of less than 0.7 and $FEV_1$ less than 80% predicted (Global Initiative for Chronic Obstructive Lung Disease spirometric stages 2–4). No individual with alternative pathologic lung diagnosis other than emphysema was included as a subject or control. Control subjects were defined by $FEV_1$/FVC ratio > 0.70. In pre-analysis filtering, we excluded genes with fewer than 1 count per million reads in at least 50% of subjects. We performed differential gene expression using the limma [69] and voom [70] packages from R/Bioconductor. We tested the associations of gene expression levels and COPD. We adjusted the models for age, race, sex, current smoking status, smoking pack-years, and library preparation batch. Surrogate variables were used to estimate other latent effects and were computed using the sva package. We controlled for multiple testing with a false discovery rate (FDR) of 1%.

### Datasets

We obtained pre-aligned hg38 GTEX tissue specific sequencing through dbGAP (dbGaP Accession phs000424.v8.p2) and focused on the top tissues where *SERPINA1* is expressed (Liver, Lung, Blood, Spleen, Kidney and Small Intestine). We re-analyzed publicly available lung RNA-seq data from 82 COPD individuals and 64 individuals with normal spirometry (Bioproject PRJNA245811). Sequencing runs without coverage in the long 3′UTR region of *SERPINA1* were excluded from analysis. We repeated our analysis with a larger set of lung RNA-Seq data from 376 individuals exhibiting COPD and 277 individuals with normal lung function (Lung Tissue Research Consortium generated by the NHLBI TOPMed Project (https://www.nhlbiwgs.org/). This larger RNA-Seq dataset is available with authorization (dbGaP Accession phs001662.v2.p1). Individuals without clear phenotype were excluded. We used the ENCODE HepG2 fractionation experiments (Bioproject PRJNA30709:GSE30567—SRR307915, SRR307916, SRR307928, SRR307929) and the patient-derived hepatocellular carcinoma cell line fractionation experiments (Bioproject PRJNA543441) to investigate cellular localization of SERPINA1 isoforms. To analyze *SERPINA1* long read sequencing, we used publicly available data from PacBio human liver aligned using IsoSeq. Links to this data and human blood and heart sequencing are available on https://s3.amazonaws.com/datasets.pacb.com/downloadtools.html.

### Supporting information

**S1 Fig. Sequence of proximal and distal polyadenylation sites in *SERPINA1* 3′ UTR.** A) *SERPINA1* mRNAS 3′ UTR sequence showing consensus polyA signal (yellow), cleavage site as determined by 3′ End sequencing (red) and putative G/U-rich region downstream of

cleavage site. B) Structural analysis of proximal site in *SERPINA1* mRNA 3′ UTR. We observe that both the stop codon (red) and APA site (purple) are highly accessible (i.e. have high SHAPE reactivity), allowing both these sites access to the Ribosome and cleavage machinery respectively.
(TIF)

**S2 Fig. Analysis of distal ratio with GOLD stages of COPD in LTRC data.** *SERPINA1* 3′ UTR distal ratio for disease severity as measured by GOLD stage rating of COPD (red) and non-COPD (blue) indicating an increase in distal ratio with increasing disease severity.
(TIF)

**S3 Fig. 3'UTR constructs are terminated at the expected length.** We measured 3′ end specific reads from A549 cells transfected with the nanoluciferase short or long 3'UTR construct and identified 3′ ends of endogenous and transfected *SERPINA1* constructs. As expected, we found that 3′ reads cluster at the end of the 3'UTR of the endogenously expressed *GAPDH* in both A) short and B) long *SERPINA1* 3'UTR transfected cells. C) We identified a single primary 3′ end for the short 3'UTR *SERPINA1* transcripts. D) We identified a single primary 3′ end for the long 3'UTR *SERPINA1* transcripts at the end of the 3'UTR, mainly in the SV40 polyA signal, indicating that there is no cleavage and polyadenylation from the mutated proximal polyA site.
(TIF)

**S4 Fig. Long and short 3'UTR *SERPINA1* isoforms have similar RNA stability.** A) We performed pulse chase experiments in A549 cells using ethylene uridine (EU) and click-it chemistry for labeling with biotin-azide to measure relative mRNA stability by qRT-PCR. GAPDH was stable over the 24-hour period. We observed a steep decline in both the long and short *SERPINA1* constructs, consistent with high ΔCT values, indicating similar stability for both long and short 3′ UTRs. B) We measured the levels of short and long *SERPINA1* 3'UTR isoforms transfected into A540 cells at equimolar amounts for steady-state RNA levels. We used two different qRT-PCR primer pairs (blue and red). We found that the short and long transcripts were present at similar levels.
(TIF)

**S5 Fig. Cluster definitions for single cell RNA-seq data.** A) tSNE (t-distributed stochastic neighbor embedding) clustering of single-cell RNA-seq from five healthy human liver biopsies identifying 20 clusters as determined by [53]. B) Lung tissue from five healthy donors UMAP (Uniform Manifold Approximation and Projection) analysis of single cell RNA-seq data reveals 25 cell type Clusters as determined by [54]. C) *SERPINA1* expression (normalized transcript counts) in liver cells is indicated by the yellow-green heatmap and reveals the highest counts in Hepatocyte cells, where six separate clusters were identified (clusters 1,3,5,6,14, and 15 indicated on cell map). D) *SERPINA1* mRNA in lung cells is predominantly expressed in Alveolar Type 1 and 2 cells as well as Macrophages. E) Liver distal ratio distributions of *SERPINA1* mRNA in single cells (indicated as open dots) for hepatocyte clusters 1,3, and 6, ab-Tcells and Plasma cells illustrates significant variation within and among cell type clusters, F) Lung distal ratio distributions in single cells indicated as dots for Alveolar Type 1 and 2, Dendritic and Macrophage cell types also indicate significant variation.
(TIF)

**S6 Fig. Post-transcriptional RNA regulation of murine *Serpina1* paralogs is not analogous to regulation of human *SERPINA1*.** A) Alignment of human *SERPINA1* and the six mouse *Serpina1* paralogs (*a-f*) around the human proximal APA site indicating that this initial APA

site is conserved in all murine *Serpina1s*. B) Alignment of human *SERPINA1* and the six mouse *Serpina1* paralogs (*a-f*) around the human QKI binding site illustrating poor conservation of the long 3'UTR. There are no canonical QKI bipartite or primary single QKI binding sites in the entire 3' region (stop codon + 2kB) of murine *Serpina1a-f*. C) Human *SERPINA1* and murine *Serpina1a-f* all contain a proximal APA site. Only *Serpina1c*, *Serpina1d* and *Serpina1f* have distal APA sites and these are not strictly conserved with the human *SERPINA1* distal site. D) and E) Isoform specific alignment of *Serpina1a-f* from two different murine tissue datasets indicates that *Serpina1a-e* transcripts are highest expressed in liver tissue while *Serpina1f* transcripts are primarily expressed in kidney tissue. F) and G) We analyzed distal reads in *Serpina1a-f* in liver, and kidney tissue where available, and found no evidence for distal reads in the 3' regions of murine *Serpina1a-f*. We calculated the median distal ratio to be less than 0.001 suggesting that mice do not express a long isoform of *Serpina1*.
(TIF)

**S1 Table. RNA binding proteins identified through eCLIP as binding the 3'UTR of *SERPINA1***
(XLSX)

**S2 Table. RBPs with significant shRNA distal ratio change and corresponding differential expression in COPD from LTRC data.**
(XLSX)

**S3 Table. Primers used for qRT-PCR and cloning.**
(XLSX)

**S1 Text. Supplemental Methods.** Additional methods used to generate and analyze data for supplemental figures and tables.
(DOCX)

## Acknowledgments

We gratefully acknowledge the studies and participants who provided biological samples to LTRC and all the individuals participating in TOPMed associated cohorts.

## Author Contributions

**Conceptualization:** Lela Lackey, Vijay Shankar, Silvia B. V. Ramos, Edwin K. Silverman, Victor E. Ortega, Michael H. Cho, Craig P. Hersh, Brian D. Hobbs, Peter Castaldi, Alain Laederach.

**Data curation:** Lela Lackey, Auyon J. Ghosh, John Platig, Zhonghui Xu, Peter Castaldi, Alain Laederach.

**Formal analysis:** Lela Lackey, Aaztli Coria, Phil Grayeski, Abigail Hatfield, Vijay Shankar, John Platig, Zhonghui Xu, Alain Laederach.

**Funding acquisition:** Lela Lackey, Alain Laederach.

**Investigation:** Lela Lackey, Aaztli Coria, Auyon J. Ghosh, Phil Grayeski, Abigail Hatfield, Vijay Shankar, John Platig, Alain Laederach.

**Project administration:** Lela Lackey, Alain Laederach.

**Resources:** Lela Lackey, Alain Laederach.

**Supervision:** Lela Lackey, Peter Castaldi, Alain Laederach.

**Validation:** Lela Lackey, Alain Laederach.

**Visualization:** Lela Lackey, Aaztli Coria, Phil Grayeski, Vijay Shankar, John Platig, Zhonghui Xu, Alain Laederach.

**Writing – original draft:** Lela Lackey, Alain Laederach.

**Writing – review & editing:** Lela Lackey, Aaztli Coria, Auyon J. Ghosh, Phil Grayeski, Abigail Hatfield, Vijay Shankar, John Platig, Zhonghui Xu, Silvia B. V. Ramos, Edwin K. Silverman, Victor E. Ortega, Michael H. Cho, Craig P. Hersh, Brian D. Hobbs, Peter Castaldi, Alain Laederach.

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
