## [Decision Letter · Decision Letter 0]

29 Jun 2021

Dear Dr Lackey,

Thank you very much for submitting your Research Article entitled 'Alternative poly-adenylation modulates α1-antitrypsin expression in chronic obstructive pulmonary disease' to PLOS Genetics.

The manuscript was fully evaluated at the editorial level and by independent peer reviewers. The reviewers have found that the manuscript has limited novelty and presents largely correlative results. They have raised issues regarding the interpretation of the RNA decay data as well as the function of QKI in the expression of alpha1-antitripsin isoforms. Based on the reviews, we will not be able to accept this version of the manuscript, but we would be willing to review a much-revised version, in which experimental evidence is provided for the specific points just mentioned. We cannot, of course, promise publication at that time.

If you decide to revise the manuscript for further consideration at PLOS Genetics, please aim to resubmit within the next 60 days, unless it will take extra time to address the concerns of the reviewers, in which case we would appreciate an expected resubmission date by email to plosgenetics@plos.org.

[LINK]

We are sorry that we cannot be more positive about your manuscript at this stage. Please do not hesitate to contact us if you have any concerns or questions.

Yours sincerely,

Mihaela Zavolan

Associate Editor

PLOS Genetics

Gregory Barsh

Editor-in-Chief

PLOS Genetics

Reviewer's Responses to Questions

**Comments to the Authors:**

Reviewer #1: The authors focus on the SERPINA1 gene, which is clinically relevant in the lung. Using computational analysis of several datasets and some experiments, the authors show that SERPINA1 is alternatively polyadenylated, which has not been reported before, that the long isoform is preferentially expressed in COPD disease, that the longer isoform leads to a reduced protein production in a reporter assay, and that various RNA binding proteins (RBPs) bind the long 3’UTR isoform, and can be predicted to be potentially relevant to the differential protein production and/or to the expression changes in COPD. Overall, the topic is of interest to the genetics community, as there are still not too many examples of interesting instances of alternative polyadenylation, and overall, both the computational and the experimental parts are performed rigorously. However, the overall conceptual advance here is modest. I do have several concerns related to over-interpretation of the presented data, and suggestions for experiments that would make the paper more suitable for PLoS Genetics.

Major comments:

1. The analysis showing that the 3’UTR length does not affect RNA levels or stability is not convincing. The authors show a metabolic lablelling chase experiment for 24 hr only, after which most labelled RNA is gone, and present it as evidence for no effect of the 3’UTR on RNA stability. Even if the half-life of the RNA is affected by 2- or 4-fold (e.g., from 2hr to 8hr), it would probably not be visible in this experiment. Several relevant time points should be tested to provide a quantitative estimate of the half -life for the short and the long UTR, also controlling for the different plasmid size used. The authors should measure also whether there is a change in steady-state mRNA levels.

2. The suggestion that QKI helps retain the SERPINA1 mRNA in the nucleus is very speculative. QKI has many possible functions, and only one of them is retention in the nucleus. The fact that the long isoform of SERPINA1 is preferentially nuclear is interesting, but can be affected by many other factors except QKI. Can the authors knock QKI down and measure subcellular localization of SERPINA1 mRNA? Otherwise, the statement should be appropriately toned down.

3. In the results presented in Figure 5, the authors reach quite strong conclusions based on overall limited data. The ENCODE shRNA data is impressive, but it only covers a limited set of RBPs, which are knocked-down to varying degrees and not completely eliminated, and the data are from liver HepG2 cells, whereas the COPD data is for lung tissue. Therefore, it is possible that there are RBPs with a stronger effect that were either not profiled by ENCODE, not sufficiently knocked down, or have a different behavior in lung cells. The conclusions of some buffering based on the number of RBPs that pass some arbitrary thresholds are over-reaching. The text should be adjusted accordingly. Since QKI appears as relevant in multiple analyses, can the authors knock it down and see how it affects their luciferase reporter results? Or the endogenous protein levels, if possible?

Minor comments:

1. Figure 3E – does deletion of Region1 affect the usage of the proximal site? In general, for each of the deletions, the authors should attempt to quantify the relative Distal/Proximal usage in the 3’UTR of their reporter.

2. Figure 4A – “Region 1” is not labelled in the figure. Also, please explain the color coding of the clusters. Are these all the proteins with clusters in the ENCODE data in this region? What threshold was used to select them?

3. Since in the single-cell data the authors do not observe any statistically significant changes, and as they indicate the overall power of the analysis is severely compromised by data sparseness, I don’t see what this part adds to the manuscript. It can be relocated to the supplement and mentioned in the discussion section.

Reviewer #2: In the study led by Lackey and colleagues, the authors continue their investigation of SERPINA1 in COPD. They demonstrate that SERPINA1 pre-mRNA is subject to APA and that this appears to change in the disease setting. They characterize this event and reveal that the long isoform is significantly less translated and retained in the nucleus. Various NGS analyses nominate QKI as a potential regulator of this process and ultimately show that SERPINA1 polyA site choice varies between cell types using single cell RNA-seq. Overall, the study is interesting and has novel aspects, but more work would be needed to recommend further consideration.

There are two significant weaknesses with this study that limit its impact and confound the interpretation. I note that both concerns are related to each other. First, the authors rely too heavily on retrospective analyses and need to conduct more experiments that directly test hypotheses. There is too much reliance on correlation in patient and consortium databases. Second, the model of QKI action on SERPINA1 is not made clear throughout the text likely because of my initial concern – more experimentation is required to clarify the model. Is it that QKI does not impact SERPINA1 polyA site choice but rather binds to the long isoform while in the nucleus thereby retaining it and preventing it from getting translated? Or is it that QKI binds the long 3’UTR while getting transcribed and effects PAS choice? Could both mechanisms be occurring? These are just some of the unanswered questions that lead one to get confused as to what exactly is happening in this disease.

Specific concerns are below:

1. The type ‘M, S, or Z alleles’ are not explained in the text. A reader not familiar with this literature would have no basis to understand their identity.

2. There is a conceptual misconnect associated between analyzing HepG2 cells and COPD lung tissues. If the authors are going to use HepG2 cells as a cellular model to focus on, then shouldn’t they be then more focused on analysis of cirrhosis in the liver in patients?

3. The two nano-luciferase reporters need to be confirmed that the correct PAS is being utilized. A straightforward 3’RACE should suffice. As this is a key finding in the paper, such confirmation would be essential.

4. The authors datamine eCLIP data to generate a hypothesis that QKI is a trans acting factor potentially regulating translation of SERPINA1. The go on to retrospectively analyze fractionated RNA-seq data and to analyze SHAPE data to ultimately conclude that QKI is a likely an important trans factor that binds to the long 3’UTR isoform of SERPINA1 mRNA and retain it in the nucleus. While these data appear convincing, they are also completely correlative. The authors need to conduct two sets of experiments where they: 1) mutate the QKI consensus site in their nano-luc reporter to determine whether there is an effect on translation and 2) knockdown QKI to determine whether SERPINA1 long isoforms are now relocalized to the cytoplasm. Without these experiments or similar types of experiments, it is impossible to understand cause/effect of QKI to SERPINA1.

5. Based upon the data presented in Figure 5A, I am puzzled how the authors make their conclusion in the context of their previous conclusions ‘QKI decreased expression in COPD therefore decreased the long isoform of the SERPINA1 mRNA, which would lead to higher translation efficiency of the A1AT protein’. This conclusion now suggests that QKI impacts PAS choice and that in the absence of QKI would result in more short isoform being produced to increase translation efficiency of A1AT protein. Or is it that the authors believe that when QKI expression goes down, the long isoform is less retained in the nucleus and is now free to increase translation?

Reviewer #3: The manuscript entitled “Alternative poly-adenylation modulates α1-antitrypsin expression in chronic obstructive pulmonary disease” described that there is an increase in the use of a distal poly-adenylation site in primary lung tissue RNA-seq in COPD cases when compared to controls. The authors also reported that the alternative polyadenylation event involves two sites, a proximal and distal site downstream of the A1AT stop codon. They measured the distal ratio in human primary tissue short read RNA-seq data and corroborated their data results with long read RNA-seq data demonstrated a 50-fold decreased translation efficiency and A1AT expression. They also identified QKI as a specific modulator of SERPINA1 mRNA translation through nuclear retention in COPD primary lung tissue. This study revealed a complex post-transcriptional mechanism that regulates alternative polyadenylation and A1AT expression in COPD and acts as a buffering mechanism for changes in cellular populations in the COPD. This is an interesting and well designed study. The findings are important and novel.

Comments:

1. In Figure 2, the authors made conclusions about the M, S and Z alleles. More information on the N numbers should be provided and better justified. In addition for the ZZ sample, how conclusions are made with N=1 as a control?

2. In Figures 3B and C it is not clear how the data from the two isoforms validate the authors’ conclusions by inhibiting translation, while showing not effect on mRNA stability? The conclusions reported here should be better supported by the data.

3. In Figure 4 the role and function of the identified RNA binding proteins to modulate translation efficiency in COPD model should be shown.

4. Moreover, in Figure 5 the data from the single cell analysis should be validated in both mRNA and protein levels to support the authors’ conclusions.

5. Finally, the manuscript lacks experiments to assess the function in a COPD model to validate the proposed mechanisms. This is a major limitation.

**Have all data underlying the figures and results presented in the manuscript been provided?**

Reviewer #1: Yes

Reviewer #2: Yes

Reviewer #3: Yes

PLOS authors have the option to publish the peer review history of their article (what does this mean?). If published, this will include your full peer review and any attached files.

Reviewer #1: No

Reviewer #2: No

Reviewer #3: No

---

## [Decision Letter · Decision Letter 1]

25 Oct 2021

Dear Dr Lackey,

We are pleased to inform you that your manuscript entitled "Alternative poly-adenylation modulates α1-antitrypsin expression in chronic obstructive pulmonary disease" has been editorially accepted for publication in PLOS Genetics. Congratulations!

Yours sincerely,

Mihaela Zavolan

Associate Editor

PLOS Genetics

Gregory Barsh

Editor-in-Chief

PLOS Genetics

Comments from the reviewers (if applicable):

Reviewer's Responses to Questions

**Comments to the Authors:**

Reviewer #1: The authors have addressed all my comments from the previous review in a satisfactory manner. I can now recommend publication in PLoS Genetics

Reviewer #2: I commend the authors on conducting a few of the key experiments that I suggested. I also believe they have done a better job muting the discussion on mechanism as the work is too preliminary to suggest a specific mode of action here. Thus, I am supportive of the manuscript in its current form.

Reviewer #3: The authors have addressed my previous comments.

**Have all data underlying the figures and results presented in the manuscript been provided?**

Reviewer #1: Yes

Reviewer #2: Yes

Reviewer #3: Yes

PLOS authors have the option to publish the peer review history of their article (what does this mean?). If published, this will include your full peer review and any attached files.

Reviewer #1: **Yes: **Igor Ulitsky

Reviewer #2: No

Reviewer #3: **Yes: **Andriana Margariti

**Data Deposition**

http://datadryad.org/submit?journalID=pgenetics&manu=PGENETICS-D-21-00741R1

**Press Queries**

---

## [Editor Report · Acceptance letter]

11 Nov 2021

PGENETICS-D-21-00741R1 

Alternative poly-adenylation modulates α1-antitrypsin expression in chronic obstructive pulmonary disease 

Dear Dr Lackey, 

We are pleased to inform you that your manuscript entitled "Alternative poly-adenylation modulates α1-antitrypsin expression in chronic obstructive pulmonary disease" has been formally accepted for publication in PLOS Genetics! Your manuscript is now with our production department and you will be notified of the publication date in due course.

With kind regards,

Agnes Pap

PLOS Genetics

On behalf of:
